# Adversarial Surrogate Losses for Ordinal Regression

**Rizal Fathony**     **Mohammad Bashiri**     **Brian D. Ziebart**
Department of Computer Science
University of Illinois at Chicago
Chicago, IL 60607
{rfatho2, mbashi4, bziebart}@uic.edu

## Abstract

Ordinal regression seeks class label predictions when the penalty incurred for mistakes increases according to an ordering over the labels. The absolute error is a canonical example. Many existing methods for this task reduce to binary classification problems and employ surrogate losses, such as the hinge loss. We instead derive uniquely defined surrogate ordinal regression loss functions by seeking the predictor that is robust to the worst-case approximations of training data labels, subject to matching certain provided training data statistics. We demonstrate the advantages of our approach over other surrogate losses based on hinge loss approximations using UCI ordinal prediction tasks.

## 1   Introduction

For many classification tasks, the discrete class labels being predicted have an inherent order (e.g., *poor*, *fair*, *good*, *very good*, and *excellent* labels). Confusing two classes that are distant from one another (e.g., *poor* instead of *excellent*) is more detrimental than confusing two classes that are nearby. The absolute error, $|\hat{y} - y|$ between label prediction ($\hat{y} \in \mathcal{Y}$) and actual label ($y \in \mathcal{Y}$) is a canonical ordinal regression loss function. The **ordinal regression** task seeks class label predictions for new datapoints that minimize losses of this kind.

Many prevalent methods reduce the ordinal regression task to subtasks solved using existing supervised learning techniques. Some view the task from the regression perspective and learn both a linear regression function and a set of thresholds that define class boundaries [1–5]. Other methods take a classification perspective and use tools from cost-sensitive classification [6–8]. However, since the absolute error of a predictor on training data is typically a non-convex (and non-continuous) function of the predictor's parameters for each of these formulations, surrogate losses that approximate the absolute error must be optimized instead. Under both perspectives, surrogate losses for ordinal regression are constructed by transforming the surrogate losses for binary zero-one loss problems—such as the hinge loss, the logistic loss, and the exponential loss—to take into account the different penalties of the ordinal regression problem. Empirical evaluations have compared the appropriateness of different surrogate losses, but these still leave the possibility of undiscovered surrogates that align better with the ordinal regression loss.

To address these limitations, we seek the most robust [9] ordinal regression predictions by focusing on the following adversarial formulation of the ordinal regression task: *what predictor best minimizes absolute error in the worst case given partial knowledge of the conditional label distribution?* We answer this question by considering the Nash equilibrium for a game defined by combining the loss function with Lagrangian potential functions [10]. We derive a surrogate loss function for empirical risk minimization that realizes this same adversarial predictor. We show that different types of available knowledge about the conditional label distribution lead to thresholded regression-based predictions or classification-based predictions. In both cases, the surrogate loss is novel compared to existing surrogate losses. We also show that our surrogate losses enjoy Fisher consistency, a desirable

theoretical property guaranteeing that minimizing the surrogate loss produces Bayes optimal decisions for the original loss in the limit. We develop two different approaches for optimizing the loss: a stochastic optimization of the primal objective and a quadratic program formulation of the dual objective. The second approach enables us to efficiently employ the kernel trick to provide a richer feature representation without an overly burdensome time complexity. We demonstrate the benefits of our adversarial formulation over previous ordinal regression methods based on hinge loss for a range of prediction tasks using UCI datasets.

## 2 Background and Related Work

### 2.1 Ordinal Regression Problems

Ordinal regression is a discrete label prediction problem characterized by an ordered penalty for making mistakes: $\text{loss}(\hat{y}_1, y) < \text{loss}(\hat{y}_2, y)$ if $y < \hat{y}_1 < \hat{y}_2$ or $y > \hat{y}_1 > \hat{y}_2$. Though many loss functions possess this property, the absolute error $|\hat{y} - y|$ is the most widely studied. We similarly restrict our consideration to this loss function in this paper. The full loss matrix $\mathbf{L}$ for absolute error with four labels is shown in Table 1. The expected loss incurred using a probabilistic predictor $\hat{P}(\hat{y}|\mathbf{x})$ evaluated on true data

Table 1: Ordinal regression loss matrix.

$$\begin{bmatrix} 0 & 1 & 2 & 3 \\ 1 & 0 & 1 & 2 \\ 2 & 1 & 0 & 1 \\ 3 & 2 & 1 & 0 \end{bmatrix}$$

distribution $P(\mathbf{x}, y)$ is: $\mathbb{E}_{\mathbf{X}, Y \sim P; \hat{Y} | \mathbf{X} \sim \hat{P}}[\mathbf{L}_{\hat{Y}, Y}] = \sum_{\mathbf{x}, y, \hat{y}} P(\mathbf{x}, y) \hat{P}(\hat{y}|\mathbf{x}) \mathbf{L}_{\hat{y}, y}$. The supervised learning objective for this problem setting is to construct a probabilistic predictor $\hat{P}(\hat{y}|\mathbf{x})$ in a way that minimizes this expected loss using training samples distributed according to the empirical distribution $\tilde{P}(\mathbf{x}, y)$, which are drawn from the unknown true data generating distribution, $P(\mathbf{x}, y)$.

A naïve ordinal regression approach relaxes the task to a continuous prediction problem, minimizes the least absolute deviation [11], and then rounds predictions to nearest integral label [12]. More sophisticated methods range from using a cumulative link model [13] that assumes the cumulative conditional probability $P(Y \leq j | \mathbf{x})$ follows a link function, to Bayesian non-parametric approaches [14] and many others [15–22]. We narrow our focus over this broad range of methods found in the related work to those that can be viewed as empirical risk minimization methods with piece-wise convex surrogates, which are more closely related to our approach.

### 2.2 Threshold Methods for Ordinal Regression

Threshold methods are one popular family of techniques that treat the ordinal response variable, $\hat{f} \triangleq \mathbf{w} \cdot \mathbf{x}$, as a continuous real-valued variable and introduce $|\mathcal{Y}| - 1$ thresholds $\theta_1, \theta_2, ..., \theta_{|\mathcal{Y}|-1}$ that partition the real line into $|\mathcal{Y}|$ segments: $\theta_0 = -\infty < \theta_1 < \theta_2 < ... < \theta_{|\mathcal{Y}|-1} < \theta_{|\mathcal{Y}|} = \infty$ [4]. Each segment corresponds to a label with $\hat{y}_i$ assigned label $j$ if $\theta_{j-1} < \hat{f} \leq \theta_j$. There are two different approaches for constructing surrogate losses based on the threshold methods to optimize the choice of $\mathbf{w}$ and $\theta_1, \ldots, \theta_{|\mathcal{Y}|-1}$: one is based on penalizing all thresholds involved when a mistake is made and one is based on only penalizing the most immediate thresholds.

*All thresholds* methods penalize every erroneous threshold using a surrogate loss, $\delta$, for sets of binary classification problems: $\text{loss}_{\text{AT}}(\hat{f}, y) = \sum_{k=1}^{y-1} \delta(-(\theta_k - \hat{f})) + \sum_{k=y}^{|\mathcal{Y}|} \delta(\theta_k - \hat{f})$. Shashua and Levin [1] studied the hinge loss under the name of *support vector machines with a sum-of margin strategy*, while Chu and Keerthi [2] proposed a similar approach under the name of *support vector ordinal regression with implicit constraints* (SVORIM). Lin and Li [3] proposed *ordinal regression boosting*, an all thresholds method using the exponential loss as a surrogate. Finally, Rennie and Srebro [4] proposed a unifying approach for all threshold methods under a variety of surrogate losses.

Rather than penalizing all erroneous thresholds when an error is made, *immediate thresholds* methods only penalize the threshold of the true label and the threshold immediately beneath the true label: $\text{loss}_{\text{IT}}(\hat{f}, y) = \delta(-(\theta_{y-1} - \hat{f})) + \delta(\theta_y - \hat{f})$.[1] Similar to the all thresholds methods, immediate threshold methods have also been studied in the literature under different names. For hinge loss surrogates, Shashua and Levin [1] called the model *support vector with fixed-margin strategy* while Chu and Keerthi [2] use the term *support vector ordinal regression with explicit constraints* (SVOREX). For

the exponential loss, Lin and Li [3] introduced *ordinal regression boosting with left-right margins*. Rennie and Srebro [4] also proposed a unifying framework for immediate threshold methods.

## 2.3 Reduction Framework from Ordinal Regression to Binary Classification

Li and Lin [5] proposed a reduction framework to convert ordinal regression problems to binary classification problems by extending training examples. For each training sample $(\mathbf{x}, y)$, the reduction framework creates $|\mathcal{Y}| - 1$ extended samples $(\mathbf{x}^{(j)}, y^{(j)})$ and assigns weight $w_{y,j}$ to each extended sample. The binary label associated with the extended sample is equivalent to the answer of the question: "is the rank of $\mathbf{x}$ greater than $j$?" The reduction framework allows a choice for how extended samples $\mathbf{x}^{(j)}$ are constructed from original samples $\mathbf{x}$ and how to perform binary classification. If the threshold method is used to construct the extended sample and SVM is used as the binary classification algorithm, the classifier can be obtained by solving a family of quadratic optimization problems that includes SVORIM and SVOREX as special instances.

## 2.4 Cost-sensitive Classification Methods for Ordinal Regression

Rather than using thresholding or the reduction framework, ordinal regression can also be cast as a special case of cost-sensitive multiclass classification. Two of the most popular classification-based ordinal regression techniques are extensions of one-versus-one (OVO) and one-versus-all (OVA) cost-sensitive classification [6, 7]. Both algorithms leverage a transformation that converts a cost-sensitive classification problem to a set of weighted binary classification problems. Rather than reducing to binary classification, Tu and Lin [8] reduce cost-sensitive classification to one-sided regression (OSR), which can be viewed as an extension of the one-versus-all (OVA) technique.

## 2.5 Adversarial Prediction

Foundational results establish a duality between adversarial logarithmic loss minimization and constrained maximization of the entropy [23]. This takes the form of a zero-sum game between a predictor seeking to minimize expected logarithmic loss and an adversary seeking to maximize this same loss. Additionally, the adversary is constrained to choose a distribution that matches certain sample statistics. Ultimately, through the duality to maximum entropy, this is equivalent to maximum likelihood estimation of probability distributions that are members of the exponential family [23]. Grünwald and Dawid [9] emphasize this formulation as a justification for the principle of maximum entropy [24] and generalize the adversarial formulation to other loss functions. Extensions to multivariate performance measures [25] and non-IID settings [26] have demonstrated the versatility of this perspective.

Recent analysis [27, 28] has shown that for the special case of zero-one loss classification, this adversarial formulation is equivalent to empirical risk minimization with a surrogate loss function:

$$\mathrm{AL}_{\mathbf{f}}^{\text{0-1}}(\mathbf{x}_i, y_i) = \max_{\mathcal{S} \subseteq \{1,\dots,|\mathcal{Y}|\}, \mathcal{S} \neq \emptyset} \sum_{j \in \mathcal{S}} (\psi_{j,y_i}(\mathbf{x}_i) + |\mathcal{S}| - 1)/|\mathcal{S}|, \tag{1}$$

where $\psi_{j,y_i}(\mathbf{x}_i)$ is the potential difference $\psi_{j,y_i}(\mathbf{x}_i) = f_j(\mathbf{x}_i) - f_{y_i}(\mathbf{x}_i)$. This surrogate loss function provides a key theoretical advantage compared to the Crammer-Singer hinge loss surrogate for multiclass classification [29]: it guarantees Fisher consistency [27] while Crammer-Singer—despite its popularity in many applications, such as Structured SVM [30, 31]—does not [32, 33]. We extend this type of analysis to the ordinal regression setting with the absolute error as the loss function in this paper, producing novel surrogate loss functions that provide better predictions than other convex, piece-wise linear surrogates.

# 3 Adversarial Ordinal Regression

## 3.1 Formulation as a zero-sum game

We seek the ordinal regression predictor that is the most robust to uncertainty given partial knowledge of the evaluating distribution's characteristics. This takes the form of a zero-sum game between a predictor player choosing a predicted label distribution $\hat{P}(\hat{y}|\mathbf{x})$ that minimizes loss and an adversarial

player choosing an evaluation distribution $\check{P}(\check{y}|\mathbf{x})$ that maximizes loss while closely matching the feature-based statistics of the training data:

$$\min_{\hat{P}(\hat{y}|\mathbf{x})} \max_{\check{P}(\check{y}|\mathbf{x})} \mathbb{E}_{\mathbf{X}\sim P;\hat{Y}|\mathbf{X}\sim\hat{P};\check{Y}|\mathbf{X}\sim\check{P}} \left[\left|\hat{Y} - \check{Y}\right|\right] \text{ such that: } \mathbb{E}_{\mathbf{X}\sim P;\check{Y}|\mathbf{X}\sim\check{P}}[\phi(\mathbf{X}, \check{Y})] = \tilde{\phi}. \quad (2)$$

The vector of feature moments, $\tilde{\phi} = \mathbb{E}_{\mathbf{X},Y\sim\tilde{P}}[\phi(\mathbf{X}, Y)]$, is measured from sample training data distributed according to the empirical distribution $\tilde{P}(\mathbf{x}, y)$.

An ordinal regression problem can be viewed as a cost-sensitive loss with the entries of the cost matrix defined by the absolute loss between the row and column labels (an example of the cost matrix for the case of a problem with four labels is shown in Table 1). Following the construction of adversarial prediction games for cost-sensitive classification [10], the optimization of Eq. (2) reduces to minimizing the equilibrium game values of a new set of zero-sum games characterized by matrix $\mathbf{L}'_{\mathbf{x}_i,\mathbf{w}}$:

$$\min_{\mathbf{w}} \sum_i \overbrace{\max_{\check{\mathbf{p}}_{\mathbf{x}_i}} \min_{\hat{\mathbf{p}}_{\mathbf{x}_i}} \hat{\mathbf{p}}_{\mathbf{x}_i}^T \mathbf{L}'_{\mathbf{x}_i,\mathbf{w}} \check{\mathbf{p}}_{\mathbf{x}_i}}^{\text{zero-sum game}}; \; \mathbf{L}'_{\mathbf{x}_i,\mathbf{w}} = \begin{bmatrix} f_1 - f_{y_i} & \cdots & f_{|\mathcal{Y}|} - f_{y_i} + |\mathcal{Y}| - 1 \\ f_1 - f_{y_i} + 1 & \cdots & f_{|\mathcal{Y}|} - f_{y_i} + |\mathcal{Y}| - 2 \\ \vdots & \ddots & \vdots \\ f_1 - f_{y_i} + |\mathcal{Y}| - 1 & \cdots & f_{|\mathcal{Y}|} - f_{y_i} \end{bmatrix}, \quad (3)$$
$$\underbrace{\phantom{\min_{\mathbf{w}} \sum_i \max_{\check{\mathbf{p}}_{\mathbf{x}_i}} \min_{\hat{\mathbf{p}}_{\mathbf{x}_i}} \hat{\mathbf{p}}_{\mathbf{x}_i}^T \mathbf{L}'_{\mathbf{x}_i,\mathbf{w}} \check{\mathbf{p}}_{\mathbf{x}_i}}}_{\text{convex optimization of } \mathbf{w}}$$

where: $\mathbf{w}$ represents a vector of Lagrangian model parameters; $f_j = \mathbf{w} \cdot \phi(\mathbf{x}_i, j)$ is a Lagrangian potential; $\hat{\mathbf{p}}_{\mathbf{x}_i}$ is a vector representation of the conditional label distribution, $\hat{P}(\hat{Y} = j|\mathbf{x}_i)$, i.e., $\hat{\mathbf{p}}_{\mathbf{x}_i} = [\hat{P}(\hat{Y} = 1|\mathbf{x}_i) \; \hat{P}(\hat{Y} = 2|\mathbf{x}_i) \; \ldots]^T$; and $\check{\mathbf{p}}_{\mathbf{x}_i}$ is similarly defined. The matrix $\mathbf{L}'_{\mathbf{x}_i,\mathbf{w}} = (|\hat{y} - \check{y}| + f_{\check{y}} - f_{y_i})$ is a zero-sum game matrix for each example. This optimization problem (Eq. (3)) is convex in $\mathbf{w}$ and the inner zero-sum game can be solved using a linear program [10]. To address finite sample estimation errors, the difference between expected and sample feature can be bounded in Eq. (2), $||\mathbb{E}_{X\sim P;\check{Y}|\mathbf{X}\sim\check{P}}[\phi(\mathbf{X}, \check{Y})] - \tilde{\phi}|| \leq \epsilon$, leading to Lagrangian parameter regularization in Eq. (3) [34].

## 3.2 Feature representations

We consider two feature representations corresponding to different training data summaries:

$$\phi_{th}(\mathbf{x}, y) = \begin{pmatrix} y\mathbf{x} \\ I(y \leq 1) \\ I(y \leq 2) \\ \vdots \\ I(y \leq |\mathcal{Y}| - 1) \end{pmatrix}; \text{ and } \quad \phi_{mc}(\mathbf{x}, y) = \begin{pmatrix} I(y = 1)\mathbf{x} \\ I(y = 2)\mathbf{x} \\ I(y = 3)\mathbf{x} \\ \vdots \\ I(y = |\mathcal{Y}|)\mathbf{x} \end{pmatrix}. \quad (4)$$

The first, which we call the **thresholded regression representation**, has size $m + |\mathcal{Y}| - 1$, where $m$ is the dimension of our input space. It induces a single shared vector of feature weights and a set of thresholds. If we denote the weight vector associated with the $y\mathbf{x}$ term as $\mathbf{w}$ and the terms associated with each sum of class indicator functions as $\theta_1, \theta_2, \ldots, \theta_{|\mathcal{Y}|-1}$, then thresholds for switching between class $j$ and $j + 1$ (ignoring other classes) occur when $\mathbf{w} \cdot \mathbf{x}_i = \theta_j$.

The second feature representation, $\phi_{mc}$, which we call the **multiclass representation**, has size $m|\mathcal{Y}|$ and can be equivalently interpreted as inducing a set of class-specific feature weights, $f_j = \mathbf{w}_j \cdot \mathbf{x}_i$. This feature representation is useful when ordered labels cannot be thresholded according to any single direction in the input space, as shown in the example dataset of Figure 1.

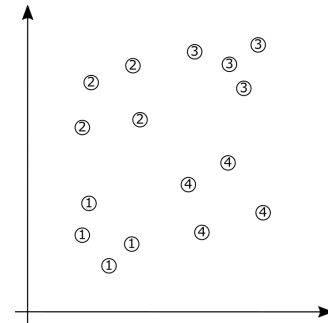

Figure 1: Example where multiple weight vectors are useful.

### 3.3 Adversarial Loss from the Nash Equilibrium

We now present the main technical contribution of our paper: a surrogate loss function that, when minimized, produces a solution to the adversarial ordinal regression problem of Eq. (3).[2]

**Theorem 1.** *An adversarial ordinal regression predictor is obtained by choosing parameters* $\mathbf{w}$ *that minimize the empirical risk of the surrogate loss function:*

$$AL_{\mathbf{w}}^{ord}(\mathbf{x}_i, y_i) = \max_{j,l \in \{1,\dots,|\mathcal{Y}|\}} \frac{f_j + f_l + j - l}{2} - f_{y_i} = \max_j \frac{f_j + j}{2} + \max_l \frac{f_l - l}{2} - f_{y_i}, \quad (5)$$

*where* $f_j = \mathbf{w} \cdot \phi(\mathbf{x}_i, j)$ *for all* $j \in \{1, \dots, |\mathcal{Y}|\}$.

*Proof sketch.* Let $j^*, l^*$ be the solution of $\text{argmax}_{j,l \in \{1,\dots,|\mathcal{Y}|\}} \frac{f_j + f_l + j - l}{2}$, we show that the Nash equilibrium value of a game matrix that contains only row $j^*$ and $l^*$ and column $j^*$ and $l^*$ from matrix $\mathbf{L}'_{\mathbf{x}_i, \mathbf{w}}$ is exactly $\frac{f_{j^*,} + f_{l^*} + j^* - l^*}{2}$. We then show that adding other rows and columns in $\mathbf{L}'_{\mathbf{x}_i, \mathbf{w}}$ to the game matrix does not change the game value. Given the resulting closed form solution of the game (instead of a minimax), we can recast the adversarial framework for ordinal regression as an empirical risk minimization with the proposed loss. □

We note that the $AL_{\mathbf{w}}^{ord}$ surrogate is the maximization over pairs of different potential functions associated with each class (including pairs of identical class labels) added to the distance between the pair. For both of our feature representations, we make use of the fact that maximization over each element of the pair can be independently realized, as shown on the right-hand side of Eq. (5).

**Thresholded regression surrogate loss**

In the thresholded regression feature representation, the parameter contains a single shared vector of feature weights $\mathbf{w}$ and $|\mathcal{Y}| - 1$ terms $\theta_k$ associated with thresholds. Following Eq. (5), the adversarial ordinal regression surrogate loss for this feature representation can be written as:

$$AL^{ord-th}(\mathbf{x}_i, y_i) = \max_j \frac{j(\mathbf{w} \cdot \mathbf{x}_i + 1) + \sum_{k \geq j} \theta_k}{2} + \max_l \frac{l(\mathbf{w} \cdot \mathbf{x}_i - 1) + \sum_{k \geq l} \theta_k}{2} - y_i \mathbf{w} \cdot \mathbf{x}_i - \sum_{k \geq y_i} \theta_k.$$

(6)

This loss has a straight-forward interpretation in terms of the thresholded regression perspective, as shown in Figure 2: it is based on averaging the thresholded label predictions for potentials $\mathbf{w} \cdot \mathbf{x}_i + 1$ and $\mathbf{w} \cdot \mathbf{x}_i - 1$. This penalization of the pair of thresholds differs from the thresholded surrogate losses of related work, which either penalize all violated thresholds or penalize only the thresholds adjacent to the actual class label.

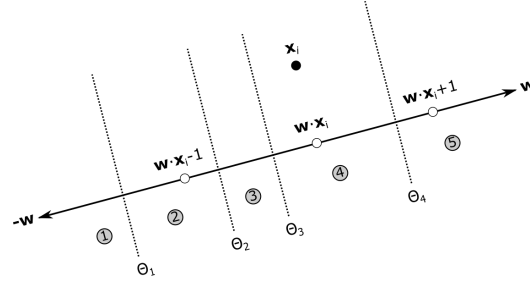

Figure 2: Surrogate loss calculation for datapoint $\mathbf{x}_i$ (projected to $\mathbf{w} \cdot \mathbf{x}_i$) with a label prediction of 4 for predictive purposes, the surrogate loss is instead obtained using potentials for the classes based on $\mathbf{w} \cdot \mathbf{x}_i + 1$ (label 5) and $\mathbf{w} \cdot \mathbf{x}_i - 1$ (label 2) averaged together.

Using a binary search procedure over $\theta_1, \dots, \theta_{|\mathcal{Y}|-1}$, the largest lower bounding threshold for each of these potentials can be obtained in $\mathcal{O}(\log |\mathcal{Y}|)$ time.

**Multiclass ordinal surrogate loss**

In the multiclass feature representation, we have a set of specific feature weights $\mathbf{w}_j$ for each label and the adversarial multiclass ordinal surrogate loss can be written as:

$$AL^{ord-mc}(\mathbf{x}_i, y_i) = \max_{j,l \in \{1,\dots,|\mathcal{Y}|\}} \frac{\mathbf{w}_j \cdot \mathbf{x}_i + \mathbf{w}_l \cdot \mathbf{x}_i + j - l}{2} - \mathbf{w}_{y_i} \cdot \mathbf{x}_i. \quad (7)$$

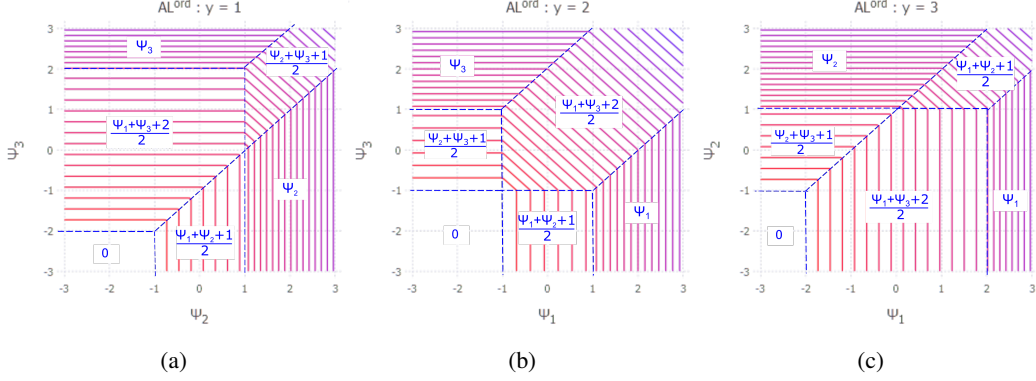

Figure 3: Loss function contour plots of $\text{AL}^{\text{ord}}$ over the space of potential differences $\psi_j \triangleq f_j - f_{y_i}$ for the prediction task with three classes when the true label is $y_i = 1$ (a), $y_i = 2$ (b), and $y_i = 3$ (c).

We can also view this as the maximization over $|\mathcal{Y}|(|\mathcal{Y}| + 1)/2$ linear hyperplanes. For an ordinal regression problem with three classes, the loss has six facets with different shapes for each true label value, as shown in Figure 3. In contrast with $\text{AL}^{\text{ord-th}}$, the class label potentials for $\text{AL}^{\text{ord-mc}}$ may differ from one another in more-or-less arbitrary ways. Thus, searching for the maximal $j$ and $l$ class labels requires $\mathcal{O}(|\mathcal{Y}|)$ time.

## 3.4 Consistency Properties

The behavior of a prediction method in ideal learning settings—i.e., trained on the true evaluation distribution and given an arbitrarily rich feature representation, or, equivalently, considering the space of all measurable functions—provides a useful theoretical validation. Fisher consistency requires that the prediction model yields the Bayes optimal decision boundary [32, 33, 35] in this setting. Given the true label conditional probability $P_j(\mathbf{x}) \triangleq P(Y = j | \mathbf{x})$, a surrogate loss function $\delta$ is said to be Fisher consistent with respect to the loss $\ell$ if the minimizer $\mathbf{f}^*$ of the surrogate loss achieves the Bayes optimal risk, i.e.,:

$$\mathbf{f}^* = \operatorname*{argmin}_{\mathbf{f}} \mathbb{E}_{Y|\mathbf{X} \sim P} \left[ \delta_{\mathbf{f}}(\mathbf{X}, Y) | \mathbf{X} = \mathbf{x} \right] \tag{8}$$
$$\Rightarrow \mathbb{E}_{Y|\mathbf{X} \sim P} \left[ \ell_{\mathbf{f}^*}(\mathbf{X}, Y) | \mathbf{X} = \mathbf{x} \right] = \min_{\mathbf{f}} \mathbb{E}_{Y|\mathbf{X} \sim P} \left[ \ell_{\mathbf{f}}(\mathbf{X}, Y) | \mathbf{X} = \mathbf{x} \right].$$

Ramaswamy and Agarwal [36] provide a necessary and sufficient condition for a surrogate loss to be Fisher consistent with respect to general multiclass losses, which includes ordinal regression losses. A recent analysis by Pedregosa et al. [35] shows that the *all thresholds* and the *immediate thresholds* methods are Fisher consistent provided that the base binary surrogates losses they use are convex with a negative derivative at zero.

For our proposed approach, the condition for Fisher consistency above is equivalent to:

$$\mathbf{f}^* = \operatorname*{argmin}_{\mathbf{f}} \sum_y P_y \left[ \max_{j,l} \frac{f_j + f_l + j - l}{2} - f_y \right] \Rightarrow \operatorname*{argmax}_j f_j^*(\mathbf{x}) \subseteq \operatorname*{argmin}_j \sum_y P_y |j - y|. \tag{9}$$

Since adding a constant to all $f_j$ does not change the value of both $\text{AL}_{\mathbf{f}}^{\text{ord}}$ and $\operatorname{argmax}_j f_j(\mathbf{x})$, we employ the constraint $\max_j f_j(\mathbf{x}) = 0$, to remove redundant solutions for the consistency analysis. We establish an important property of the minimizer for $\text{AL}_{\mathbf{f}}^{\text{ord}}$ in the following theorem.

**Theorem 2.** *The minimizer vector* $\mathbf{f}^*$ *of* $\mathbb{E}_{Y|\mathbf{X} \sim P} \left[ AL_{\mathbf{f}}^{ord}(\mathbf{X}, Y) | \mathbf{X} = \mathbf{x} \right]$ *satisfies the* loss reflective *property, i.e., it complements the absolute error by starting with a negative integer value, then increasing by one until reaching zero, and then incrementally decreases again.*

*Proof sketch.* We show that for any $\mathbf{f}^0$ that does not satisfy the loss reflective property, we can construct $\mathbf{f}^1$ using several steps that satisfy the loss reflective property and has the expected loss value less than the expected loss of $\mathbf{f}^0$. □

Example vectors $\mathbf{f}^*$ that satisfy Theorem 2 are $[0, -1, -2]^{\mathrm{T}}$, $[-1, 0, -1]^{\mathrm{T}}$ and $[-2, -1, 0]^{\mathrm{T}}$ for three-class problems, and $[-3, -2, -1, 0, -1]$ for five-class problems. Using the key property of the minimizer above, we establish the consistency of our loss functions in the following theorem.

**Theorem 3.** *The adversarial ordinal regression surrogate loss $AL^{ord}$ from Eq. (5) is Fisher consistent.*

*Proof sketch.* We only consider $|\mathcal{Y}|$ possible values of $\mathbf{f}$ that satisfy the loss reflective property. For the $\mathbf{f}$ that corresponds to class $j$, the value of the expected loss is equal to the Bayes loss if we predict $j$ as the label. Therefore minimizing over $\mathbf{f}$ that satisfy the loss reflective property is equivalent to finding the Bayes optimal response. $\square$

### 3.5 Optimization

#### 3.5.1 Primal Optimization

To optimize the regularized adversarial ordinal regression loss from the primal, we employ stochastic average gradient (SAG) methods [37, 38], which have been shown to converge faster than standard stochastic gradient optimization. The idea of SAG is to use the gradient of each example from the last iteration where it was selected to take a step. However, the naïve implementation of SAG requires storing the gradient of each sample, which may be expensive in terms of the memory requirements. Fortunately, for our loss $AL_{\mathbf{w}}^{ord}$, we can drastically reduce this memory requirement by just storing a pair of number, $(j^*, l^*) = \mathrm{argmax}_{j,l \in \{1,\dots,|\mathcal{Y}|\}} \frac{f_j + f_l + j - l}{2}$, rather than storing the gradient for each sample. Appendix C explains the details of this technique.

#### 3.5.2 Dual Optimization

Dual optimization is often preferred when optimizing piecewise linear losses, such as the hinge loss, since it enables one to easily perform the kernel trick and obtain a non-linear decision boundary without heavily sacrificing computational efficiency. Optimizing the regularized adversarial ordinal regression loss in the dual can be performed by solving the following quadratic optimization:

$$\max_{\boldsymbol{\alpha}, \boldsymbol{\beta}} \sum_{i,j} j(\alpha_{i,j} - \beta_{i,j}) - \frac{1}{2} \sum_{i,j,k,l} (\alpha_{i,j} + \beta_{i,j})(\alpha_{k,l} + \beta_{k,l})(\phi(\mathbf{x}_i, j) - \phi(\mathbf{x}_i, y_i)) \cdot (\phi(\mathbf{x}_k, l) - \phi(\mathbf{x}_l, y_k))$$

subject to: $\alpha_{i,j} \geq 0; \beta_{i,j} \geq 0; \sum_j \alpha_{i,j} = \frac{C}{2}; \sum_j \beta_{i,j} = \frac{C}{2}; i, k \in \{1, \dots, n\}; j, l \in \{1, \dots, |\mathcal{Y}|\}.$ (10)

Note that our dual formulation only depends on the dot product of the features. Therefore, we can also easily apply the kernel trick to our algorithm. Appendix D describes the derivation from the primal optimization to the dual optimization above.

## 4 Experiments

### 4.1 Experiment Setup

We conduct our experiments on a benchmark dataset for ordinal regression [14], evaluate the performance using mean absolute error (MAE), and perform statistical tests on the results of different hinge loss surrogate methods. The benchmark contains datasets taken from the UCI Machine Learning repository [39] ranging from relatively small to relatively large in size. The characteristics of the datasets, including the number of classes, the training set size, the testing set size, and the number of features, are described in Table 2.

Table 2: Dataset properties.

| Dataset | #class | #train | #test | #features |
|---|---|---|---|---|
| diabetes | 5 | 30 | 13 | 2 |
| pyrimidines | 5 | 51 | 23 | 27 |
| triazines | 5 | 130 | 56 | 60 |
| wisconsin | 5 | 135 | 59 | 32 |
| machinecpu | 10 | 146 | 63 | 6 |
| autompg | 10 | 274 | 118 | 7 |
| boston | 5 | 354 | 152 | 13 |
| stocks | 5 | 665 | 285 | 9 |
| abalone | 10 | 2923 | 1254 | 10 |
| bank | 10 | 5734 | 2458 | 8 |
| computer | 10 | 5734 | 2458 | 21 |
| calhousing | 10 | 14447 | 6193 | 8 |

In the experiment, we consider different methods using the original feature space and a kernelized feature space using the Gaussian radial basis function kernel. The methods that we compare include two variations of our approach, the threshold based (AL^{ord-th}), and the multiclass-based (AL^{ord-mc}).

Table 3: The average of the mean absolute error (MAE) for each model. Bold numbers in each case indicate that the result is the best or not significantly worse than the best (paired t-test with $\alpha = 0.05$).

| Dataset | Threshold-based models | | | | Multiclass-based models | | | | |
|---|---|---|---|---|---|---|---|---|---|
| | AL[ord-th] | RED[th] | AT | IT | AL[ord-mc] | RED[mc] | CSOSR | CSOVO | CSOVA |
| diabetes | **0.696** | **0.715** | 0.731 | 0.827 | **0.629** | **0.700** | **0.715** | 0.738 | 0.762 |
| pyrimidines | 0.654 | 0.678 | 0.615 | 0.626 | **0.509** | 0.565 | **0.520** | 0.576 | **0.526** |
| triazines | **0.607** | 0.683 | 0.649 | 0.654 | 0.670 | 0.673 | 0.677 | 0.738 | 0.732 |
| wisconsin | **1.077** | **1.067** | **1.097** | 1.175 | 1.136 | 1.141 | 1.208 | 1.275 | 1.338 |
| machinecpu | **0.449** | **0.456** | **0.458** | **0.467** | 0.518 | 0.515 | 0.646 | 0.602 | 0.702 |
| autompg | **0.551** | **0.550** | **0.550** | 0.617 | 0.599 | 0.602 | 0.741 | 0.598 | 0.731 |
| boston | 0.316 | **0.304** | **0.306** | **0.298** | 0.311 | 0.311 | 0.353 | **0.294** | 0.363 |
| stocks | 0.324 | 0.317 | 0.315 | 0.324 | 0.168 | 0.175 | 0.204 | **0.147** | 0.213 |
| abalone | 0.551 | 0.547 | 0.546 | 0.571 | **0.521** | **0.520** | 0.545 | 0.558 | 0.556 |
| bank | 0.461 | 0.460 | 0.461 | 0.461 | **0.445** | **0.446** | 0.732 | 0.448 | 0.989 |
| computer | 0.640 | 0.635 | 0.633 | 0.683 | **0.625** | **0.624** | 0.889 | 0.649 | 1.055 |
| calhousing | 1.190 | 1.183 | 1.182 | 1.225 | 1.164 | **1.144** | 1.237 | 1.202 | 1.601 |
| average | 0.626 | 0.633 | 0.629 | 0.661 | 0.613 | 0.618 | 0.706 | 0.652 | 0.797 |
| # bold | 5 | 5 | 4 | 2 | 5 | 5 | 2 | 2 | 1 |

The baselines we use for the threshold-based models include a SVM-based reduction framework algorithm (RED[th]) [5], an all threshold method with hinge loss (AT) [1, 2], and an immediate threshold method with hinge loss (IT) [1, 2]. For the multiclass-based models, we compare our method with an SVM-based reduction algorithm using multiclass features (RED[mc]) [5], with cost-sensitive one-sided support vector regression (CSOSR) [8], with cost-sensitive one-versus-one SVM (CSOVO) [7], and with cost-sensitive one-versus-all SVM (CSOVA) [6]. For our Gaussian kernel experiment, we compare our threshold based model (AL[ord-th]) with SVORIM and SVOREX [2].

In our experiments, we first make 20 random splits of each dataset into training and testing sets. We performed two stages of five-fold cross validation on the first split training set for tuning each model's regularization constant $\lambda$. In the first stage, the possible values for $\lambda$ are $2^{-i}, i = \{1, 3, 5, 7, 9, 11, 13\}$. Using the best $\lambda$ in the first stage, we set the possible values for $\lambda$ in the second stage as $2^{\frac{i}{2}}\lambda_0, i = \{-3, -2, -1, 0, 1, 2, 3\}$, where $\lambda_0$ is the best parameter obtained in the first stage. Using the selected parameter from the second stage, we train each model on the 20 training sets and evaluate the MAE performance on the corresponding testing set. We then perform a statistical test to find whether the performance of a model is different with statistical significance from other models. We perform the Gaussian kernel experiment similarly with model parameter $C$ equals to $2^i, i = \{0, 3, 6, 9, 12\}$ and kernel parameter $\gamma$ equals to $2^i, i = \{-12, -9, -6, -3, 0\}$ in the first stage. In the second stage, we set $C$ equals to $2^i C_0, i = \{-2, -1, 0, 1, 2\}$ and $\gamma$ equals to $2^i \gamma_0, i = \{-2, -1, 0, 1, 2\}$, where $C_0$ and $\gamma_0$ are the best parameters obtained in the first stage.

## 4.2 Results

We report the mean absolute error (MAE) averaged over the dataset splits as shown in Table 3 and Table 4. We highlight the result that is either the best or not worse than the best with statistical significance (under paired t-test with $\alpha = 0.05$) in boldface font. We also provide the summary for each model in terms of the averaged MAE over all datasets and the number of datasets for which each model marked with boldface font in the bottom of the table.

As we can see from Table 3, in the experiment with the original feature space, threshold-based models perform well on relatively small datasets, whereas multiclass-based models perform well on relatively large datasets. A possible explanation for this result is that multiclass-based models have more flexibility in creating decision boundaries, hence they perform better if the training data size is sufficient. However, since multiclass-based models have many more parameters than threshold-based models ($m|\mathcal{Y}|$ parameters rather than $m + |\mathcal{Y}| - 1$ parameters), multiclass methods may need more data, and hence, may not perform well on relatively small datasets.

In the threshold-based models comparison, AL[ord-th], RED[th], and AT perform competitively on relatively small datasets like triazines, wisconsin, machinecpu, and autompg. AL[ord-th] has a

slight advantage over RED[th] on the overall accuracy, and a slight advantage over AT on the number of "indistinguishably best" performance on all datasets. We can also see that AT is superior to IT in the experiments under the original feature space.

Among the multiclass-based models, AL[ord-mc] and RED[mc] perform competitively on datasets like `abalone`, `bank`, and `computer`, with a slight advantage of AL[ord-mc] model on the overall accuracy. In general, the cost-sensitive models perform poorly compared with AL[ord-mc] and RED[mc]. A notable exception is the CSOVO model which perform very well on the `stocks` and the `boston` datasets.

In the Gaussian kernel experiment, we can see from Table 4 that the kernelized version of AL[ord-th] performs significantly better than the threshold-based models SVORIM and SVOREX in terms of both the overall accuracy and the number of "indistinguishably best" performance

Table 4: The average of MAE for models with Gaussian kernel.

| Dataset | AL[ord-th] | SVORIM | SVOREX |
|---|---|---|---|
| diabetes | **0.696** | **0.665** | **0688** |
| pyrimidines | **0.478** | 0.539 | 0.550 |
| triazines | **0.609** | **0.612** | **0.604** |
| wisconsin | 1.090 | 1.113 | **1.049** |
| machinecpu | **0.452** | 0.652 | 0.628 |
| autompg | **0.529** | 0.589 | 0.593 |
| boston | **0.278** | 0.324 | 0.316 |
| stocks | **0.103** | **0.099** | **0.100** |
| average | 0.531 | 0.574 | 0.566 |
| # bold | 7 | 3 | 4 |

on all datasets. We also note that immediate-threshold-based model (SVOREX) performs better than all-threshold-based model (SVORIM) in our experiment using Gaussian kernel. We can conclude that our proposed adversarial losses for ordinal regression perform competitively compared to the state-of-the-art ordinal regression models using both original feature spaces and kernel feature spaces with a significant performance improvement in the Gaussian kernel experiments.

## 5 Conclusion and Future Work

In this paper, we have proposed a novel surrogate loss for ordinal regression, a classification problem where the discrete class labels have an inherent order and penalty for making mistakes based on that order. We focused on the absolute loss, which is the most widely used ordinal regression loss. In contrast with existing methods, which typically reduce ordinal regression to binary classification problems and then employ surrogates for the binary zero-one loss, we derive a unique surrogate ordinal regression loss by seeking the predictor that is robust to a worst case constrained approximation of the training data. We derived two versions of the loss based on two different feature representation approaches: thresholded regression and multiclass representations. We demonstrated the benefit of our approach on a benchmark of datasets for ordinal regression tasks. Our approach performs competitively compared to the state-of-the-art surrogate losses based on hinge loss. We also demonstrated cases when the multiclass feature representations works better than thresholded regression representation, and vice-versa, in our experiments.

Our future work will investigate less prevalent ordinal regression losses, such as the discrete quadratic loss and arbitrary losses that have v-shaped penalties. Furthermore, we plan to investigate the characteristics required of discrete ordinal losses for their optimization to have a compact analytical solution. In terms of applications, one possible direction of future work is to combine our approach with deep neural network models to perform end-to-end representation learning for ordinal regression applications like age estimation and rating prediction. In that setting, our proposed loss can be used in the last layer of a deep neural network to serve as the gradient source for the backpropagation algorithm.

## Acknowledgments

This research was supported as part of the Future of Life Institute (futureoflife.org) FLI-RFP-AI1 program, grant#2016-158710 and by NSF grant RI-#1526379.

## Footnotes

[1]For the boundary labels, the method defines $\delta(-(\theta_0 - \hat{f})) = \delta(\theta_{y+1} - \hat{f}) = 0$.

[2]The detailed proof of this theorem and others are contained in the supplementary materials. Proof sketches are presented in the main paper.

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
