[Supplementary Material]

# Supplementary Materials

## A  Proof for the Adversarial Ordinal Regression Loss (Theorem 1)

Before proving Theorem 1, we review the game matrix $\mathbf{L}'_{\mathbf{x}_i,\mathbf{w}}$ for ordinal regression problems. Below is the matrix when the number of classes is four:

$$
\mathbf{L}'_{\mathbf{x}_i,\mathbf{w}} = \begin{bmatrix}
f_1 - f_{y_i} & f_2 - f_{y_i} + 1 & f_3 - f_{y_i} + 2 & f_4 - f_{y_i} + 3 \\
f_1 - f_{y_i} + 1 & f_2 - f_{y_i} & f_3 - f_{y_i} + 1 & f_4 - f_{y_i} + 2 \\
f_1 - f_{y_i} + 2 & f_2 - f_{y_i} + 1 & f_3 - f_{y_i} & f_4 - f_{y_i} + 1 \\
f_1 - f_{y_i} + 3 & f_2 - f_{y_i} + 2 & f_3 - f_{y_i} + 1 & f_4 - f_{y_i}
\end{bmatrix} \tag{11}
$$

$$
= \begin{bmatrix}
f_1 & f_2 + 1 & f_3 + 2 & f_4 + 3 \\
f_1 + 1 & f_2 & f_3 + 1 & f_4 + 2 \\
f_1 + 2 & f_2 + 1 & f_3 & f_4 + 1 \\
f_1 + 3 & f_2 + 2 & f_3 + 1 & f_4
\end{bmatrix} - f_{y_i} \tag{12}
$$

$$
= \mathbf{L}''_{\mathbf{x}_i,\mathbf{w}} - f_{y_i}. \tag{13}
$$

**Theorem 1.** *An adversarial ordinal regression predictor is obtained by choosing parameters $\mathbf{w}$ that minimize the empirical risk of the surrogate loss function:*

$$
AL^{ord}_{\mathbf{w}}(\mathbf{x}_i, y_i) = \max_{j,l \in \{1,\dots,|\mathcal{Y}|\}} \frac{f_j + f_l + j - l}{2} - f_{y_i} = \max_j \frac{f_j + j}{2} + \max_l \frac{f_l - l}{2} - f_{y_i}, \tag{14}
$$

*where $f_j = \mathbf{w} \cdot \phi(\mathbf{x}_i, j)$ for all $j \in \{1, \dots, |\mathcal{Y}|\}$.*

*Proof.* Our proof strategy is to use the inequalities implied by the definition of $AL^{ord}_{\mathbf{w}}$ and then show that the value of $AL^{ord}_{\mathbf{w}}$ is equal to the game value of sub-matrices of $\mathbf{L}'_{\mathbf{x}_i,\mathbf{w}}$. We start by showing the equality for a small 2 by 2 sub-matrix and build up until we show that the value of $AL^{ord}_{\mathbf{w}}$ is indeed equal to the game value of the whole game matrix $\mathbf{L}'_{\mathbf{x}_i,\mathbf{w}}$. Empirically minimizing $AL^{ord}_{\mathbf{w}}$ will conclude the theorem.

Let us begin the proof by denoting $v(\mathbf{G})$ as the Nash equilibrium value of a game characterized by game matrix $\mathbf{G}$. We would like to prove that for a zero-sum game characterized by $\mathbf{L}'_{\mathbf{x}_i,\mathbf{w}}$ as described in Eq. (3), $v(\mathbf{L}'_{\mathbf{x}_i,\mathbf{w}}) = \max_{j,l \in \{1,\dots,|\mathcal{Y}|\}} \frac{f_j + f_l + j - l}{2} - f_{y_i}$.

Note that for any game matrix $\mathbf{G}$ and any constant $c$, $v(\mathbf{G} + c) = v(\mathbf{G}) + c$. We denote $\mathbf{L}''_{\mathbf{x}_i,\mathbf{w}} = \mathbf{L}'_{\mathbf{x}_i,\mathbf{w}} + f_{y_i}$. Thus, proving the theorem is equivalent to proving $v(\mathbf{L}''_{\mathbf{x}_i,\mathbf{w}}) = \max_{j,l \in \{1,\dots,|\mathcal{Y}|\}} \frac{f_j + f_l + j - l}{2}$. The matrix $\mathbf{L}''_{\mathbf{x}_i,\mathbf{w}}$ is similar to the matrix in Eq. (3), but without including the $-f_{y_i}$ term in each its cells, i.e.,

$$
\mathbf{L}''_{\mathbf{x}_i,\mathbf{w}} = \begin{bmatrix}
f_1 & f_2 + 1 & \cdots & f_{|\mathcal{Y}|-1} + |\mathcal{Y}| - 2 & f_{|\mathcal{Y}|} + |\mathcal{Y}| - 1 \\
f_1 + 1 & f_2 & \cdots & f_{|\mathcal{Y}|-1} + |\mathcal{Y}| - 3 & f_{|\mathcal{Y}|} + |\mathcal{Y}| - 2 \\
\vdots & \vdots & \ddots & \vdots & \vdots \\
f_1 + |\mathcal{Y}| - 2 & f_2 + |\mathcal{Y}| - 3 & \cdots & f_{|\mathcal{Y}|-1} & f_{|\mathcal{Y}|} + 1 \\
f_1 + |\mathcal{Y}| - 1 & f_2 + |\mathcal{Y}| - 2 & \cdots & f_{|\mathcal{Y}|-1} + 1 & f_{|\mathcal{Y}|}
\end{bmatrix}. \tag{15}
$$

Let $j^*$ and $l^*$ be the solution of $\operatorname{argmax}_{j,l \in \{1,\dots,|\mathcal{Y}|\}} \frac{f_j + f_l + j - l}{2}$ (if there are ties, pick any of them) and let $u^* = \max_{j,l \in \{1,\dots,|\mathcal{Y}|\}} \frac{f_j + f_l + j - l}{2} = \frac{f_{j^*} + f_{l^*} + j^* - l^*}{2}$. We know the following inequalities hold:

$$
f_{j^*} + f_{l^*} + j^* - l^* \geq f_j + f_l + j - l, \quad \forall j, l \in \{1, \dots, |\mathcal{Y}|\} \tag{16}
$$

$$
f_{j^*} + j^* \geq f_j + j, \quad \forall j \in \{1, \dots, |\mathcal{Y}|\} \tag{17}
$$

$$
f_{l^*} - l^* \geq f_l - l, \quad \forall l \in \{1, \dots, |\mathcal{Y}|\}. \tag{18}
$$

We also know that $j^* \geq l^*$; otherwise, we could just swap them to obtain a larger value.

We first focus on the cases where $j^* \neq l^*$. We analyze three different games that are characterized by subsets of matrix $\mathbf{L}''_{\mathbf{x}_i,\mathbf{w}}$ and show that the value of those games is $u^*$.

**Case 1:** Let $\mathbf{G}_1$ be a game characterized by a 2 by 2 matrix with values that are taken from rows and columns $j^*$ and $l^*$ of matrix $\mathbf{L}''_{\mathbf{x}_i, \mathbf{w}}$, i.e.,

$$\mathbf{G}_1 = \begin{bmatrix} f_{l^*} & f_{j^*} + j^* - l^* \\ f_{l^*} + j^* - l^* & f_{j^*} \end{bmatrix}. \tag{19}$$

We will show that $v(\mathbf{G}_1) = u^*$. Let $\mathbf{p}$ be the vector of adversary's mixed strategy, then finding $v(\mathbf{G}_1)$ is equivalent with solving the following optimization:

$$\max V \tag{20}$$
$$\text{s.t. } V \le p_{l^*} f_{l^*} + p_{j^*}(f_{j^*} + j^* - l^*) = p_{l^*} f_{l^*} + p_{j^*} f_{j^*} + p_{j^*}(j^* - l^*)$$
$$V \le p_{l^*}(f_{l^*} + j^* - l^*) + p_{j^*} f_{j^*} = p_{l^*} f_{l^*} + p_{j^*} f_{j^*} + p_{l^*}(j^* - l^*).$$

We now analyze the optimization above. Let $p_{l^*} = 0.5 - \alpha$ and $p_{j^*} = 0.5 + \alpha$ for some $\alpha$ where $-0.5 \le \alpha \le 0.5$. The optimization above become:

$$\max V \tag{21}$$
$$\text{s.t. } V \le (0.5 - \alpha) f_{l^*} + (0.5 + \alpha) f_{j^*} + (0.5 + \alpha)(j^* - l^*)$$
$$= 0.5\,(f_{l^*} + f_{j^*} + j^* - l^*) + \alpha\,[(f_{j^*} - f_{l^*}) + (j^* - l^*)]$$
$$V \le (0.5 - \alpha) f_{l^*} + (0.5 + \alpha) f_{j^*} + (0.5 - \alpha)(j^* - l^*)$$
$$= 0.5\,(f_{l^*} + f_{j^*} + j^* - l^*) + \alpha\,[(f_{j^*} - f_{l^*}) - (j^* - l^*)].$$

Since $j^* \ne l^*$, based on Eq. (16), we know that:

$$f_{j^*} + f_{l^*} + j^* - l^* \ge f_{j^*} + f_{j^*} + j^* - j^* \Leftrightarrow (f_{j^*} - f_{l^*}) - (j^* - l^*) \le 0, \tag{22}$$
$$f_{j^*} + f_{l^*} + j^* - l^* \ge f_{l^*} + f_{l^*} + l^* - l^* \Leftrightarrow (f_{j^*} - f_{l^*}) + (j^* - l^*) \ge 0. \tag{23}$$

Therefore, the optimal solution is to set $\alpha = 0$, since setting nonzero $\alpha$ will decrease the right-hand side of one of the constraints and hence decrease the value of $V$. Thus, the solution is achieved when we set $p_{l^*} = p_{j^*} = 0.5$, which results in a game value of $\frac{f_{j^*} + f_{l^*} + j^* - l^*}{2} = u^*$.[3]

**Case 2:** Let $\mathbf{G}_2$ be a game characterized by a $|\mathcal{Y}|$ by 2 matrix with values that are taken from column $j^*$ and $l^*$ of matrix $\mathbf{L}''_{\mathbf{x}_i, \mathbf{w}}$, i.e.,

$$\mathbf{G}_2 = \begin{bmatrix} f_{l^*} + l^* - 1 & f_{j^*} + j^* - 1 \\ \vdots & \vdots \\ f_{l^*} & f_{j^*} + j^* - l^* \\ f_{l^*} + 1 & f_{j^*} + j^* - l^* - 1 \\ \vdots & \vdots \\ f_{l^*} + j^* - l^* - 1 & f_{j^*} + 1 \\ f_{l^*} + j^* - l^* & f_{j^*} \\ \vdots & \vdots \\ f_{l^*} + |\mathcal{Y}| - l^* & f_{j^*} + |\mathcal{Y}| - j^* \end{bmatrix}. \tag{24}$$

Finding $v(\mathbf{G}_2)$ is equivalent to solving a similar optimization to that of Eq (20) with $|\mathcal{Y}|$ constraints corresponding to each row of matrix $\mathbf{G}_2$ instead of just two. We can easily see that the solution is achieved if we set $p_{l^*} = p_{j^*} = 0.5$ as in the previous case. The right hand side of any $m$-th constraint $m < l^*$ or $m > j^*$ is dominated, i.e., it has value greater than or equal to $u^*$, and the right hand side of any $m$-th constraint $l^* < m < l^*$ is equal to $u^*$. Assigning other values to $p_{l^*}$ and $p_{j^*}$ will decrease the right-hand side of some of the $m$-th ($l^* \le m \le j^*$) constraints (as explained in case 1), and hence decrease the value of $V$. Therefore, we can conclude that $v(\mathbf{G}_2) = u^*$.

**Case 3:** Let $\mathbf{G}_3$ be a game characterized by a $|\mathcal{Y}|$ by 3 matrix with values that are taken from columns $j^*$, $l^*$, and any other column $m$ in matrix $\mathbf{L}''_{\mathbf{x}_i, \mathbf{w}}$. We consider three variations of the game, $\mathbf{G}_3^1$ where

$m < l^*$, $\mathbf{G}_3^2$ where $l^* < m < j^*$, and $\mathbf{G}_3^3$ where $m > j^*$. Below is the game matrix for the first variation:

$$\mathbf{G}_3^1 = \begin{bmatrix} \vdots & \vdots & \vdots \\ f_m & f_{l^*} + l^* - m & f_{j^*} + j^* - m \\ \vdots & \vdots & \vdots \\ f_m + l^* - m & f_{l^*} & f_{j^*} + j^* - l^* \\ \vdots & \vdots & \vdots \\ f_m + j^* - m & f_{l^*} + j^* - l^* & f_{j^*} \\ \vdots & \vdots & \vdots \end{bmatrix}. \tag{25}$$

Let us analyze the optimization for finding the game value for $\mathbf{G}_3^1$, in particular the $l^*$-th and $j^*$-th constraints:

$$\max V \tag{26}$$

$$\text{s.t.} \vdots$$
$$V \le p_m(f_m + l^* - m) + p_{l^*} f_{l^*} + p_{j^*}(f_{j^*} + j^* - l^*)$$
$$V \le p_m(f_m + j^* - m) + p_{l^*}(f_{l^*} + j^* - l^*) + p_{j^*} f_{j^*}$$

$$\vdots$$

Let us use the notation similar to Case 1. Let $p_m = \beta$, $p_{l^*} = 0.5 - \alpha - \beta$ and $p_{j^*} = 0.5 + \alpha$ where $-0.5 \le \alpha \le 0.5$; $0 \le \beta \le 1$; and $-0.5 \le \alpha + \beta \le 0.5$. We can write the constraints above as:

$$V \le 0.5 \left(f_{l^*} + f_{j^*} + j^* - l^*\right) + \alpha \left[(f_{j^*} - f_{l^*}) + (j^* - l^*)\right] + \beta \left[(f_m - m) - (f_{l^*} - l^*)\right]$$
$$V \le 0.5 \left(f_{l^*} + f_{j^*} + j^* - l^*\right) + \alpha \left[(f_{j^*} - f_{l^*}) - (j^* - l^*)\right] + \beta \left[(f_m - m) - (f_{l^*} - l^*)\right].$$

Since $(f_{j^*} - f_{l^*}) + (j^* - l^*) \ge 0$; $(f_{j^*} - f_{l^*}) - (j^* - l^*) \le 0$; and $(f_m - m) - (f_{l^*} - l^*) \le 0$, the optimal solution is setting $\alpha = 0$, and $\beta = 0$. Since $p_m = \beta = 0$, we leave with the same game matrix as $\mathbf{G}_2$. Therefore $v(\mathbf{G}_3^1) = u^*$.

For $\mathbf{G}_3^3$, we let $p_m = \beta$, $p_{l^*} = 0.5 - \alpha$ and $p_{j^*} = 0.5 + \alpha - \beta$ where $-0.5 \le \alpha \le 0.5$; $0 \le \beta \le 1$; and $-0.5 \le \alpha - \beta \le 0.5$. Similar to the previous case, $l^*$-th and $j^*$-th constraints can be written as:

$$V \le 0.5 \left(f_{l^*} + f_{j^*} + j^* - l^*\right) + \alpha \left[(f_{j^*} - f_{l^*}) + (j^* - l^*)\right] + \beta \left[(f_m + m) - (f_{j^*} + j^*)\right]$$
$$V \le 0.5 \left(f_{l^*} + f_{j^*} + j^* - l^*\right) + \alpha \left[(f_{j^*} - f_{l^*}) - (j^* - l^*)\right] + \beta \left[(f_m + m) - (f_{j^*} + j^*)\right].$$

Due to a similar reason as in the previous case, and $(f_m + m) - (f_{j^*} + j^*) \le 0$, the optimal solution is to set $\alpha = 0$, and $\beta = 0$, and hence $v(\mathbf{G}_3^3) = u^*$.

For $\mathbf{G}_3^2$, we will analyze the $l^*$-th, $m$-th, and $j^*$-th constraint. Let $p_m = \beta$, $p_{l^*} = 0.5 - \alpha$ and $p_{j^*} = 0.5 + \alpha - \beta$ where $-0.5 \le \alpha \le 0.5$; $0 \le \beta \le 1$; and $-0.5 \le \alpha - \beta \le 0.5$. The constraints can be written as:

$$V \le 0.5 \left(f_{l^*} + f_{j^*} + j^* - l^*\right) + \alpha \left[(f_{j^*} - f_{l^*}) + (j^* - l^*)\right] + \beta \left[(f_m + m) - (f_{j^*} + j^*)\right]$$
$$V \le 0.5 \left(f_{l^*} + f_{j^*} + j^* - l^*\right) + \alpha \left[f_{j^*} - f_{l^*} + j^* + l^* - 2m\right] + \beta \left[(f_m + m) - (f_{j^*} + j^*)\right]$$
$$V \le 0.5 \left(f_{l^*} + f_{j^*} + j^* - l^*\right) + \alpha \left[(f_{j^*} - f_{l^*}) - (j^* - l^*)\right] + \beta \left[(f_m - m) - (f_{j^*} - j^*)\right].$$

We know that $(f_{j^*} - f_{l^*}) + (j^* - l^*) \ge 0$; $(f_{j^*} - f_{l^*}) - (j^* - l^*) \le 0$ and $(f_m + m) - (f_{j^*} + j^*) \le 0$. If it is the case that $f_{j^*} - f_{l^*} + j^* + l^* - 2m \le 0$, or $(f_m - m) - (f_{j^*} - j^*) \le 0$, or both, it will force both $\alpha$ and $\beta$ to be 0. If both of them are positive, we need an additional analysis as the following.

We focus on the $m$-th, and $j^*$-th constraints. Since we want to check if there is a combination of $\alpha$ and $\beta$ values that make the game value greater than $u^*$, $\alpha$ and $\beta$ have to satisfy the following:

$$\alpha \left[f_{j^*} - f_{l^*} + j^* + l^* - 2m\right] + \beta \left[(f_m + m) - (f_{j^*} + j^*)\right] \ge 0 \tag{27}$$

$$\Leftrightarrow \alpha \ge \frac{(f_{j^*} + j^*) - (f_m + m)}{f_{j^*} - f_{l^*} + j^* + l^* - 2m} \beta = \frac{(f_{j^*} + j^*) - (f_m - m) - 2m}{(f_{j^*} + j^*) - (f_{l^*} - l^*) - 2m} \beta \ge \beta, \tag{28}$$

$$\alpha\left[(f_{j^*} - f_{l^*}) - (j^* - l^*)\right] + \beta\left[(f_m - m) - (f_{j^*} - j^*)\right] \geq 0 \tag{29}$$

$$\Leftrightarrow \beta \geq \frac{(j^* - l^*) - (f_{j^*} - f_{l^*})}{(f_m - m) - (f_{j^*} - j^*)}\alpha = \frac{(f_{l^*} - l^*) - (f_{j^*} - j^*)}{(f_m - m) - (f_{j^*} - j^*)}\alpha \geq \alpha. \tag{30}$$

We know that $(f_{j^*}+j^*)-(f_m-m)-2m \geq (f_{j^*}+j^*)-(f_{l^*}-l^*)-2m$, and $(f_{l^*}-l^*)-(f_{j^*}-j^*) \geq (f_m - m) - (f_{j^*} - j^*)$. If at least one of those inequalities is strict, e.g., the first inequality, it is better to set $\alpha = \beta = 0$, since in order to increase the value of RHS of the $m$-th constraint $\alpha$ has to be strictly greater than $\beta$, which will decrease the RHS of the $j^*$-th constraint and thus decrease the game value. If both are equal, then many solutions exist, i.e., $\alpha = \beta$, but the game value remains the same, i.e. $u^*$, since in this case $\alpha\left[f_{j^*} - f_{l^*} + j^* + l^* - 2m\right] + \beta\left[(f_m + m) - (f_{j^*} + j^*)\right] = \alpha\left[(f_{j^*} - f_{l^*}) - (j^* - l^*)\right] + \beta\left[(f_m - m) - (f_{j^*} - j^*)\right] = 0$. Therefore $v(\mathbf{G}_3^2) = u^*$.

Note that we omit the analysis for the trivial cases when the terms associated with $\alpha$ and $\beta$ are zero. In those cases, any value of $\alpha$ and $\beta$ will satisfy the constraints, but the game value remain the same.

**Conclusion:** We are now ready to analyze the game value for $\mathbf{L}''_{\mathbf{x}_i,\mathbf{w}}$. Since adding any column $m \in \{1,\ldots,|\mathcal{Y}|\}\backslash\{l^*,j^*\}$ to $\mathbf{G}_2$ will not change the game value, then adding the combination of them will not change the game value either. Therefore, we can conclude that $v(\mathbf{L}''_{\mathbf{x}_i,\mathbf{w}}) = u^*$.

For the case that $j^* = l^*$, we know that $\max_{j,l\in\{1,\ldots,|\mathcal{Y}|\}} \frac{f_j+f_l+j-l}{2} = f_{j^*}$. It is clear that $f_{j^*}$ is the solution for the game that is defined by column $j^*$ from matrix $\mathbf{L}''_{\mathbf{x}_i,\mathbf{w}}$. For any other column $m$, if we include it in the game, the corresponding $j^*$-th constraint become (we let $p_m = \beta$, and $p_{j^*} = 1 - \beta$):

$$V \leq f_{j^*} + \beta\left[(f_m - m) - (f_{j^*} - j^*)\right] \qquad \text{if } m < j^*, \text{ or} \tag{31}$$
$$V \leq f_{j^*} + \beta\left[(f_m + m) - (f_{j^*} + j^*)\right] \qquad \text{if } m > j^*. \tag{32}$$

Since we know that $(f_{j^*} - j^*) \geq (f_m - m)$, and $(f_{j^*} + j^*) \geq (f_m + m)$, the optimal solution is to set $\beta = 0$, and the game value remain the same. We can also generalize it to all combination of column $m \in \{1,\ldots,|\mathcal{Y}|\}\backslash\{j^*\}$ to show that $v(\mathbf{L}''_{\mathbf{x}_i,\mathbf{w}}) = f_{j^*} = u^*$.

Therefore, we can conclude that the value of the game matrix $v(\mathbf{L}''_{\mathbf{x}_i,\mathbf{w}}) = \max_{j,l\in\{1,\ldots,|\mathcal{Y}|\}} \frac{f_j+f_l+j-l}{2}$, which proves the theorem. $\qquad\square$

# B  Proof in the Consistency Analysis (Theorem 2 & Theorem 3)

**Theorem 2.** *The minimizer vector $\mathbf{f}^*$ of $\mathbb{E}_{Y|\mathbf{X}\sim P}\left[AL_{\mathbf{f}}^{ord}(\mathbf{X},Y)|\mathbf{X}=\mathbf{x}\right]$ satisfies the* loss reflective *property, i.e., it complements the absolute error by starting with a negative integer value, then increasing by one until reaching zero, and then incrementally decreases again.*

*Proof.* We start the proof by analyzing the minimizer $\mathbf{f}^*$ using $P_y \triangleq P(y|\mathbf{x})$ as follows:

$$\mathbf{f}^* = \operatorname*{argmin}_{\mathbf{f}} \mathbb{E}_{Y|\mathbf{X}\sim P}\left[AL_{\mathbf{f}}^{ord}(\mathbf{X},Y)|\mathbf{X}=\mathbf{x}\right] \tag{33}$$

$$= \operatorname*{argmin}_{\mathbf{f}} \sum_y P_y \left[\max_{j,l\in\{1,\ldots,|\mathcal{Y}|\}} \frac{f_j+f_l+j-l}{2} - f_y\right] \tag{34}$$

$$= \operatorname*{argmin}_{\mathbf{f}} \left[\sum_y P_y \max_{j,l\in\{1,\ldots,|\mathcal{Y}|\}} \frac{f_j+f_l+j-l}{2} - \sum_y P_y f_y\right] \tag{35}$$

$$= \operatorname*{argmin}_{\mathbf{f}} \left[\max_{j,l\in\{1,\ldots,|\mathcal{Y}|\}} \frac{f_j+f_l+j-l}{2} - \sum_y P_y f_y\right]. \tag{36}$$

In this proof, we employ a constraint to the potential function, $\max f_j(\mathbf{x}) = 0$, in order to remove redundant solutions, as adding any constant $c$ to $\mathbf{f}$ does not change the value of both $\operatorname{argmax} f_j(\mathbf{x})$, and $\mathbb{E}_{Y|\mathbf{X}\sim P}\left[AL_{\mathbf{f}}^{ord}(\mathbf{X},Y)|\mathbf{X}=\mathbf{x}\right]$:

$$\max_{j,l\in\{1,\ldots,|\mathcal{Y}|\}} \frac{f_j+c+f_l+c+j-l}{2} - \sum_y P_y(f_y+c) \tag{37}$$

$$=c+\max_{j,l\in\{1,\ldots,|\mathcal{Y}|\}}\frac{f_j+f_l+j-l}{2}-c-\sum_y P_y(f_y)\tag{38}$$

$$=\max_{j,l\in\{1,\ldots,|\mathcal{Y}|\}}\frac{f_j+f_l+j-l}{2}-\sum_y P_y(f_y).\tag{39}$$

Let $j^*$ and $l^*$ be the solution of $\operatorname{argmax}_{j,l\in\{1,\ldots,|\mathcal{Y}|\}}\frac{f_j+f_l+j-l}{2}$. We will start from the first case where $j^* = l^*$. In this case, the minimization in Eq. (36) can be reduced to $\operatorname{argmin}_{\mathbf{f}}\left[\max_{j\in\{1,\ldots,|\mathcal{Y}|\}}f_j-\sum_y P_y f_y\right]$. Since $j^* = l^*$, we know that the following inequalities hold:

$$f_{j^*}\geq f_j\quad\forall j\in\{1,\ldots,|\mathcal{Y}|\}\tag{40}$$
$$f_{j^*}+j^*\geq f_j+j,\quad\forall j\in\{1,\ldots,|\mathcal{Y}|\}\tag{41}$$
$$f_{j^*}-j^*\geq f_j-j,\quad\forall j\in\{1,\ldots,|\mathcal{Y}|\}.\tag{42}$$

Therefore, by Eq. (40) and constraint $\max f_j(\mathbf{x})=0$, we have $f_{j^*}=0$. Then by Eq. (41), for any $i>0$, $f_{j^*+i}\leq f_{j^*}-i=-i$; and also by Eq. (42), for any $i>0$, $f_{j^*-i}\leq f_{j^*}-i=-i$. Since we want to minimize $f_{j^*}-\sum_y P_y f_y=-\sum_y P_y f_y$, the optimal solution is to set $f_{j^*+i}=-i$ and $f_{j^*-i}=-i$ for any $i>0$. Therefore we get vector $\mathbf{f}^*$ that satisfies the *loss reflective* property, i.e., it complements the absolute error by starting with a negative integer value, then increasing by one until reaching zero, and then incrementally decreases again.

We next analyze the second case where $j^*\neq l^*$. In this case, the following inequalities hold:

$$f_{j^*}+j^*\geq f_{j^*+i}+j^*+i\quad\Leftrightarrow\quad f_{j^*+i}\leq f_{j^*}-i,\quad\forall i\in\{-j^*+1,\ldots,|\mathcal{Y}|-j^*\}\tag{43}$$
$$f_{l^*}-l^*\geq f_{l^*+i}-l^*-i\quad\Leftrightarrow\quad f_{l^*+i}\leq f_{l^*}+i,\quad\forall i\in\{-l^*+1,\ldots,|\mathcal{Y}|-l^*\}.\tag{44}$$

We also know that for any $m\in\{1,\ldots,|\mathcal{Y}|\}$ the following holds:

$$m<l^*\quad\Rightarrow f_m\leq f_{l^*}-(l^*-m)\quad\text{and}\quad f_m\leq f_{j^*}+(j^*-m)\tag{45}$$
$$m>j^*\quad\Rightarrow f_m\leq f_{j^*}-(m-j^*)\quad\text{and}\quad f_m\leq f_{l^*}+(m-l^*)\tag{46}$$
$$l^*<m<j^*\quad\Rightarrow f_m\leq f_{l^*}+(m-l^*)\quad\text{and}\quad f_m\leq f_{j^*}+(j^*-m).\tag{47}$$

The relation between $f_{j^*}$ and $f_{l^*}$ in the following also holds:

$$f_{j^*}\leq f_{l^*}+j^*-l^*\tag{48}$$
$$f_{l^*}\leq f_{j^*}+j^*-l^*.\tag{49}$$

Let $\mathbf{f}^0$ be any potential function which falls into the second case (the solution of $(j^*,l^*)=\operatorname{argmax}_{j,l\in\{1,\ldots,|\mathcal{Y}|\}}\frac{f_j^0+f_l^0+j-l}{2}$ satisfies $j^*\neq l^*$) where $\mathbf{f}^0$ does not satisfy the *loss reflective* property. Let us define $h(\mathbf{f})=\frac{f_{j^*}+f_{l^*}+j^*-l^*}{2}-\sum_y P_y f_y$. We will show that we can construct $\mathbf{f}^1$ as follows. Starting from $\mathbf{f}^1=\mathbf{f}^0$ we increase all the values of $f_m^1$ for $m\in\{1,\ldots,|\mathcal{Y}|\}\setminus\{l^*,j^*\}$ such that it satisfies the constraints above with equality for the one that has minimum value. For example, in a 7-class ordinal regression where $l^*=2$ and $j^*=6$, one of possible value for $\mathbf{f}^0$ is $[-3,-1.4,-0.8,-0.2,-0.7,0,-1.2]^{\mathrm{T}}$ which satisfies all the constraints above. In this case $\mathbf{f}^1$ will be $[-2.4,-1.4,-0.4,0.6,1,0,-1]^{\mathrm{T}}$. Since the value of $\frac{f_{j^*}+f_{l^*}+j^*-l^*}{2}$ remains the same and the value of $\sum_y P_y f_y$ is increasing, we know that $h(\mathbf{f}^1)<h(\mathbf{f}^0)$. We know that in $\mathbf{f}^1$, $f_j-f_{j-1}$ is equal to 1 or -1, except for a pair $(a,b)$, where $l^*\leq a<b\leq j^*$. In the example above $a=4,b=5,f_a^1=0.6$, and $f_b^1=1$. We also know that $\frac{f_{j^*}^1+f_{l^*}^1+j^*-l^*}{2}=\frac{f_a^1+f_b^1+1}{2}$.

We now construct $\mathbf{f}^2$ from $\mathbf{f}^1$ as follows. If $\sum_{y=1}^a P_y\leq 0.5$, we set $f_j^2=f_j^1-(f_a^1-f_b^1+1)$ for $j\in\{1,\ldots,a\}$ and set $f_j^2=f_j^1$ for $j\in\{b,\ldots,|\mathcal{Y}|\}$; otherwise we set $f_j^2=f_j^1$ for $j\in\{1,\ldots,a\}$ and set $f_j^2=f_j^1-(f_b^1-f_a^1+1)$ for $j\in\{b,\ldots,|\mathcal{Y}|\}$. For the example above, if $\sum_{y=1}^a P_y\leq 0.5$ then $\mathbf{f}^2=[-3,-2,-1,0,1,0,-1]$, otherwise $\mathbf{f}^2=[-2.4,-1.4,-0.4,0.6,-0.4,-1.4,-2.4]$. We claim that $h(\mathbf{f}^2)\leq h(\mathbf{f}^1)$ as shown for the case that $\sum_{y=1}^a P_y\leq 0.5$ (the other case follows in a similar way):

$$h(\mathbf{f}^2)=\max_{j,l\in\{1,\ldots,|\mathcal{Y}|\}}\frac{f_j^2+f_l^2+j-l}{2}-\sum_y P_y f_y^2=f_b^2-\sum_y P_y f_y^2\tag{50}$$

$$= f_b^2 - \sum_{y=1}^{a} P_y f_y^2 - \sum_{y=b}^{|\mathcal{Y}|} P_y f_y^2 \tag{51}$$

$$= f_b^1 - \sum_{y=1}^{a} P_y \left[ f_y^1 - (f_a^1 - f_b^1 + 1) \right] - \sum_{y=b}^{|\mathcal{Y}|} P_y f_y^1 \tag{52}$$

$$= f_b^1 + \sum_{y=1}^{a} P_y \left[ f_a^1 - f_b^1 + 1 \right] - \sum_{y} P_y f_y^1 \tag{53}$$

$$\leq f_b^1 + 0.5 \left[ f_a^1 - f_b^1 + 1 \right] - \sum_{y} P_y f_y^1 = \frac{f_a^1 + f_b^1 + 1}{2} - \sum_{y} P_y f_y^1 = h(\mathbf{f}^1). \tag{54}$$

Finally, we construct $\mathbf{f}^3 = \mathbf{f}^2 - \max_j f_j^2$. Since adding a constant to any $\mathbf{f}$ does not change the value of $h(\mathbf{f})$, we know that $h(\mathbf{f}^3) = h(\mathbf{f}^2)$. We also know that $\mathbf{f}^3$ satisfies the *loss reflective* property described above. As an example, in the case $\sum_{y=1}^{a} P_y \leq 0.5$, then $\mathbf{f}^3 = [-4, -3, -2, -1, 0, -1, -2]$.

Since for any $\mathbf{f}^0$ that falls into the second case where the solution for $(j^*, l^*) = \operatorname{argmax}_{j,l \in \{1,...,|\mathcal{Y}|\}} \frac{f_j^0 + f_l^0 + j - l}{2}$ satisfies $j^* \neq l^*$ and $\mathbf{f}^0$ does not satisfy the *loss reflective* property, we can construct $\mathbf{f}^3$ which satisfies the *loss reflective* property and having the value of $h(\mathbf{f}^3) < h(\mathbf{f}^0)$, then $\mathbf{f}^0$ cannot be the minimizer. Therefore, we can conclude that in the first and second cases, the minimizer has to satisfy the *loss reflective* property which complete the proof of the theorem. $\square$

**Theorem 3.** *The adversarial ordinal regression surrogate loss $AL^{ord}$ from Eq. (5) is Fisher consistent.*

*Proof.* We denote $h(\mathbf{f}) \triangleq \mathbb{E}_{Y|\mathbf{X} \sim P} \left[ \text{AL}_{\mathbf{f}}^{\text{ord}}(\mathbf{X}, Y) | \mathbf{X} = \mathbf{x} \right]$. Based on Theorem 2, the minimization $\operatorname{argmin} h(\mathbf{f})$ reduces to the minimization over the set that contains all $\mathbf{f}$ that satisfies the *loss reflective* property and $\max_j f_j = 0$. Note that the set contains only $|\mathcal{Y}|$ items. In the case of $\operatorname{argmax}_j f_j = j^*$, we know that $\mathbf{f}$ that satisfies the *loss reflective* property has values $f_j = -|j^* - j|$, and hence:

$$h(\mathbf{f}) = \sum_{y} P_y \left[ \max_{j,l \in \{1,...,|\mathcal{Y}|\}} \frac{f_j + f_l + j - l}{2} - f_y \right] = \sum_{y} P_y \left[ f_{j^*} - f_y \right] = f_{j^*} - \sum_{y=1}^{|\mathcal{Y}|} P_y f_y \tag{55}$$

$$= -\sum_{y=1}^{|\mathcal{Y}|} P_y f_y = -\sum_{y=1}^{|\mathcal{Y}|} P_y \left( -|j^* - y| \right) = \sum_{y=1}^{|\mathcal{Y}|} P_y |j^* - y|.$$

Therefore, the minimizer $\mathbf{f}^* = \operatorname{argmin} h(\mathbf{f})$ satisfies $\operatorname{argmax}_j f_j^*(\mathbf{x}) \subseteq \operatorname{argmin}_j \sum_y P_y |j - y|$ and implies Fisher consistency. $\square$

## C    Primal Optimization in Details

To optimize the regularized adversarial ordinal regression loss in the primal, we employ stochastic average gradient (SAG) methods [37, 38]. SAG has been shown to converge faster than standard stochastic gradient optimization [37, 38]. In this section, we focus on the adversarial adversarial ordinal regression with multiclass representation ($\text{AL}_{\mathbf{w}}^{\text{ord-mc}}$). A version for the thresholded regression representation follows in a similar way.

Given the regularization constant $\lambda$ and the learning rate $\alpha$, the standard batch gradient update for risk minimization can be written as:

$$\mathbf{w}^{t+1} = \mathbf{w}^t - \alpha \left[ \frac{1}{n} \sum_{i=1}^{n} \mathbf{g}_i^t + \lambda \mathbf{w}^t \right] = (1 - \alpha\lambda) \mathbf{w}^t - \frac{\alpha}{n} \sum_{i=1}^{n} \mathbf{g}_i^t, \tag{56}$$

where $\mathbf{g}_i$ is the loss gradient with respect to $i$-th example. The idea of SAG is to use the gradient of each example from the last iteration where it was selected to take a step. However, the naïve implementation of SAG requires storing the gradient of each sample, which may be expensive in terms of the memory requirements.

---

**Algorithm 1** SAG for adversarial ordinal regression with multiclass representation

---

1: **Input:** training dataset with pairs $\{\mathbf{x}_i, y_i\}$, learning rate $\alpha$, regularization constant $\lambda$
2: $m \leftarrow 0$                                  {the number of sampled pairs so far}
3: $\mathbf{d} \leftarrow \mathbf{0}$                                    {for storing $\sum_{i=1}^{m} \mathbf{g}_i$}
4: $\mathrm{j}_i \leftarrow 0, \mathrm{l}_i \leftarrow 0$ for $i = 1, 2, \ldots, n$
5: **repeat**
6:      Sample $i$ from $\{1, \ldots, n\}$
7:      $j^*, l^* \leftarrow \operatorname{argmax}_{j,l} \frac{\mathbf{w}_j \cdot \mathbf{x}_i + \mathbf{w}_l \cdot \mathbf{x}_i + j - l}{2} - \mathbf{w}_{y_i} \cdot \mathbf{x}_i$
8:      **if** it is the first time we sample $i$ **then**
9:          $m \leftarrow m + 1$
10:         $\mathbf{d}_{j^*} \leftarrow \mathbf{d}_{j^*} + \frac{1}{2}\mathbf{x}_i, \ \ \mathbf{d}_{l^*} \leftarrow \mathbf{d}_{l^*} + \frac{1}{2}\mathbf{x}_i$
11:         $\mathbf{d}_{y_i} \leftarrow \mathbf{d}_{y_i} - \mathbf{x}_i$
12:      **else**
13:         $\mathbf{d}_{\mathrm{j}_i} \leftarrow \mathbf{d}_{\mathrm{j}_i} - \frac{1}{2}\mathbf{x}_i, \ \ \mathbf{d}_{\mathrm{l}_i} \leftarrow \mathbf{d}_{\mathrm{l}_i} - \frac{1}{2}\mathbf{x}_i$
14:         $\mathbf{d}_{j^*} \leftarrow \mathbf{d}_{j^*} + \frac{1}{2}\mathbf{x}_i, \ \ \mathbf{d}_{l^*} \leftarrow \mathbf{d}_{l^*} + \frac{1}{2}\mathbf{x}_i$
15:      **end if**
16:      $\mathrm{j}_i \leftarrow j^*, \mathrm{l}_i \leftarrow l^*$
17:      $\mathbf{w} \leftarrow (1 - \alpha\lambda)\mathbf{w} - \frac{\alpha}{m}\mathbf{d}$
18: **until** converge

---

Fortunately, for $\mathrm{AL}_{\mathbf{w}}^{\text{ord-mc}}$, we can drastically reduce this memory requirement by not directly storing the gradient using the following technique. Let $j^*, l^* = \operatorname{argmax}_{j,l} \frac{\mathbf{w}_j \cdot \mathbf{x}_i + \mathbf{w}_l \cdot \mathbf{x}_i + j - l}{2} - \mathbf{w}_{y_i} \cdot \mathbf{x}_i$. Assuming that $j^* \neq l^* \neq y_i$, we know that the sub-gradients are: $\nabla_{\mathbf{w}_{j^*}} = \frac{1}{2}\mathbf{x}_i$, $\nabla_{\mathbf{w}_{l^*}} = \frac{1}{2}\mathbf{x}_i$, and $\nabla_{\mathbf{w}_{y_i}} = -\mathbf{x}_i$, while $\nabla_{\mathbf{w}_k} = \mathbf{0}$ for $k \in \{1, \ldots, |\mathcal{Y}|\} \backslash \{j^*, l^*, y_i\}$. Therefore, instead of storing the sub-gradient, we can just store $j^*$ and $l^*$. Let us denote $\mathrm{j}_i$ and $\mathrm{l}_i$ for $i = 1, 2, \ldots, n$ as the storage for each example's last $j^*$ and $l^*$. We also construct a vector $\mathbf{d}$ which has the same length as our parameter vector $\mathbf{w}$ to store the sum of the latest gradients, i.e. $\mathbf{d} = \sum_{i=1}^{m} \mathbf{g}_i$, where $m$ is the number of training pairs $\{\mathbf{x}_i, y_i\}$ sampled so far. Using this notation, Algorithm 1 describes this technique for implementing SAG for adversarial ordinal regression loss with multiclass representation.

## D    Dual Optimization in Details

Based on Equation 5, the primal optimization of regularized adversarial ordinal regression loss can be written as:

$$\min_{\mathbf{w}} \ \frac{1}{2}\|\mathbf{w}\|^2 + C \sum_{i=1}^{n} \left[ \max_{j \in 1, \ldots, |\mathcal{Y}|} \frac{\mathbf{w} \cdot \phi(\mathbf{x}_i, j) + j}{2} + \max_{j \in 1, \ldots, |\mathcal{Y}|} \frac{\mathbf{w} \cdot \phi(\mathbf{x}_i, j) - j}{2} - \mathbf{w} \cdot \phi(\mathbf{x}_i, y_i) \right] \tag{57}$$

$$= \min_{\mathbf{w}} \ \frac{1}{2}\|\mathbf{w}\|^2 + \frac{C}{2} \sum_{i=1}^{n} \max_{j \in 1, \ldots, |\mathcal{Y}|} \left( \mathbf{w} \cdot \phi(\mathbf{x}_i, j) - \mathbf{w} \cdot \phi(\mathbf{x}_i, y_i) + j \right) \tag{58}$$

$$+ \frac{C}{2} \sum_{i=1}^{n} \max_{j \in 1, \ldots, |\mathcal{Y}|} \left( \mathbf{w} \cdot \phi(\mathbf{x}_i, j) - \mathbf{w} \cdot \phi(\mathbf{x}_i, y_i) - j \right).$$

The optimization above is equivalent with the following constrained optimization:

$$\min_{\mathbf{w}} \ \frac{1}{2}\|\mathbf{w}\|^2 + \frac{C}{2} \sum_{i=1}^{n} \xi_i + \frac{C}{2} \sum_{i=1}^{n} \delta_i \tag{59}$$

$$\text{subject to:} \quad \xi_i \geq \mathbf{w} \cdot \phi(\mathbf{x}_i, j) - \mathbf{w} \cdot \phi(\mathbf{x}_i, y_i) + j \qquad \forall i \in \{1, \ldots n\}; j \in \{1, \ldots, |\mathcal{Y}|\}$$

$$\delta_i \geq \mathbf{w} \cdot \phi(\mathbf{x}_i, j) - \mathbf{w} \cdot \phi(\mathbf{x}_i, y_i) - j \qquad \forall i \in \{1, \ldots n\}; j \in \{1, \ldots, |\mathcal{Y}|\}.$$

The Lagrangian for the optimization above is:

$$\mathcal{L} = \frac{1}{2}\|\mathbf{w}\|^2 + \frac{C}{2}\sum_{i=1}^{n}\xi_i + \frac{C}{2}\sum_{i=1}^{n}\delta_i - \sum_{i=1}^{n}\sum_{j=1}^{|\mathcal{Y}|}\alpha_{i,j}[\xi_i - \mathbf{w}\cdot\phi(\mathbf{x}_i,j) + \mathbf{w}\cdot\phi(\mathbf{x}_i,y_i) - j] \quad (60)$$

$$- \sum_{i=1}^{n}\sum_{j=1}^{|\mathcal{Y}|}\beta_{i,j}[\delta_i - \mathbf{w}\cdot\phi(\mathbf{x}_i,j) + \mathbf{w}\cdot\phi(\mathbf{x}_i,y_i) + j].$$

The KKT conditions:

$$\nabla_{\mathbf{w}}\mathcal{L} = \mathbf{w} - \sum_{i=1}^{n}\sum_{j=1}^{|\mathcal{Y}|}\alpha_{i,j}[-\phi(\mathbf{x}_i,j) + \phi(\mathbf{x}_i,y_i)] - \sum_{i=1}^{n}\sum_{j=1}^{|\mathcal{Y}|}\beta_{i,j}[-\phi(\mathbf{x}_i,j) + \phi(\mathbf{x}_i,y_i)] = 0$$

$$\implies \mathbf{w} = \sum_{i=1}^{n}\sum_{j=1}^{|\mathcal{Y}|}(\alpha_{i,j} + \beta_{i,j})\left[\phi(\mathbf{x}_i,y_i) - \phi(\mathbf{x}_i,j)\right]$$

$$\nabla_{\xi_i}\mathcal{L} = \frac{C}{2} - \sum_{j=1}^{|\mathcal{Y}|}\alpha_{i,j} = 0 \qquad\qquad \implies \sum_{j=1}^{|\mathcal{Y}|}\alpha_{i,j} = \frac{C}{2}$$

$$\nabla_{\delta_i}\mathcal{L} = \frac{C}{2} - \sum_{j=1}^{|\mathcal{Y}|}\beta_{i,j} = 0 \qquad\qquad \implies \sum_{j=1}^{|\mathcal{Y}|}\beta_{i,j} = \frac{C}{2}$$

$$\forall i,j,\ \alpha_{i,j}[\xi_i - \mathbf{w}\cdot\phi(\mathbf{x}_i,j) + \mathbf{w}\cdot\phi(\mathbf{x}_i,y_i) - j] = 0$$

$$\implies \alpha_{i,j} = 0 \vee \xi_i = \mathbf{w}\cdot\phi(\mathbf{x}_i,j) - \mathbf{w}\cdot\phi(\mathbf{x}_i,y_i) + j$$

$$\forall i,j,\ \beta_{i,j}[\delta_i - \mathbf{w}\cdot\phi(\mathbf{x}_i,j) + \mathbf{w}\cdot\phi(\mathbf{x}_i,y_i) + j] = 0$$

$$\implies \beta_{i,j} = 0 \vee \delta_i = \mathbf{w}\cdot\phi(\mathbf{x}_i,j) - \mathbf{w}\cdot\phi(\mathbf{x}_i,y_i) - j.$$

Rearranging the Lagrangian formula and then plugging the definition of $\mathbf{w}$ in terms of the dual variables and applying the constraints yields:

$$\mathcal{L} = \sum_{i=1}^{n}\sum_{j=1}^{|\mathcal{Y}|}j(\alpha_{i,j} - \beta_{i,j}) \qquad\qquad (61)$$

$$- \frac{1}{2}\sum_{i,k=1}^{n}\sum_{j,l=1}^{|\mathcal{Y}|}(\alpha_{i,j} + \beta_{i,j})(\alpha_{k,l} + \beta_{k,l})(\phi(\mathbf{x}_i,j) - \phi(\mathbf{x}_i,y_i))\cdot(\phi(\mathbf{x}_k,l) - \phi(\mathbf{x}_l,y_k)).$$

Therefore, the dual optimization can be written as:

$$\max_{\boldsymbol{\alpha},\boldsymbol{\beta}}\sum_{i,j}j(\alpha_{i,j} - \beta_{i,j}) \qquad\qquad (62)$$

$$- \frac{1}{2}\sum_{i,j,k,l}(\alpha_{i,j} + \beta_{i,j})(\alpha_{k,l} + \beta_{k,l})(\phi(\mathbf{x}_i,j) - \phi(\mathbf{x}_i,y_i))\cdot(\phi(\mathbf{x}_k,l) - \phi(\mathbf{x}_l,y_k))$$

subject to: $\alpha_{i,j} \geq 0; \beta_{i,j} \geq 0; \sum_j \alpha_{i,j} = \frac{C}{2}; \sum_j \beta_{i,j} = \frac{C}{2}; i,k \in \{1,\ldots,n\}; j,l \in \{1,\ldots,|\mathcal{Y}|\}.$

## Footnotes

[3] In this analysis and other analyses in this proof, we omit the analysis for the trivial cases where the terms associated with $\alpha$ (in the case above: $(f_{j^*} - f_{l^*}) + (j^* - l^*)$ and $(f_{j^*} - f_{l^*}) - (j^* - l^*)$) are zero. In this case, the value of $\alpha$ can be anything, but the game value remain the same.