[Reviews · NeurIPS 2017]

Reviewer 1



The paper proposes an adversarial approach to ordinal regression, building upon recent works along these lines for cost-sensitive losses. The proposed method is shown to be consistent, and to have favourable empirical performance compared to existing methods. The basic idea of the paper is simple yet interesting: since ordinal regression can be viewed as a type of multiclass classification, and the latter has recently been attacked by adversarial learning approaches with some success, one can combine the two to derive adversarial ordinal regression approaches. By itself this would make the contribution a little narrow, but it is further shown that the adversarial loss in this particular problem admits a tractable form (Thm 1), which allows for efficient optimisation. Fisher-consistency of the approach also follows as a consequence of existing results for the cost-sensitive case, which is a salient feature of the approach. The idea proves effective as seen in the good empirical results of the proposed method. Strictly, the performance isn't significantly better than existing methods, but rather is competitive; it would be ideal if taking the adversarial route led to improvements, but they at least illustrate that the idea is conceptually sound. Overall, I thus think the paper makes a good contribution. My only two suggestions are: o It seems prudent to give some more intuition for why the proposed adversarial approach is expected to result in a surrogate that is more appealing than standard ones (beyond automatically having consistency) -- this is mentioned in the Introduction, but could be reiterated in Sec 2.5. o As the basis of the paper is that ordinal regression with the absolute error loss can be viewed as a cost-sensitive loss, it might help to spell out concretely this connection, i.e. specify what exactly the cost matrix is. This would make transparent e.g. Eqn 3 in terms of their connection to the work of (Asif et al., 2015). Other comments: - notation E_{P(x, y) \hat{P}(\hat{y} | x)}[ L_{\hat{Y}, Y} ] is a bit confusing -- Y, \hat{Y} are presumably random variables with specified distribution? And the apparent lack of x on the RHS may confuse. - the sentence "using training samples distributed according to P̃(x, y), which are drawn from P(x, y)" is hard to understand - Sec 3.2 and elsewhere, may as well define \hat{f} = w \cdot x, not w \cdot x_i; dependence on i doesn't add much here - Certainly consistency compared to Crammer-Singer SVM is nice, but there are other multi-class formulations which are consistent, so I'm not sure this is the strongest argument for the proposed approach? - what is the relative runtime of the proposed methods in the experiments? - it wasn't immediately clear why one couldn't use the existing baselines, e.g. RED^th, in the case of a Gaussian kernel (possibly by learning with the empirical kernel map).

Reviewer 2



The paper concerns the problem of ordinal classification. The authors derive a unique surrogate ordinal regression loss by seeking the predictor that is robust to a worst case constrained approximation of the training data. The theoretical part is built upon the recent results published in "Adversarial Multiclass Classification: A Risk Minimization Perspective" (NIPS 2016). The authors could notice that the simplest approach to ordinal classification relies on estimating the conditional probabilities and then applying the Bayes rule on them (see, for example, "Ordinal Classification with Decision Rules", MCD 2007). Let me also add that ordinal classification is closely related to monotone classification or isotonic regression. Some theoretical results in this regards have been obtained in "Rule Learning with Monotonicity Constraints" (ICML, 2009) and "On Nonparametric Ordinal Classification with Monotonicity Constraints" (IEEE Transactions on Knowledge and Data Engineering, 2013). In general it is interesting paper that is worth publishing at NIPS. Minor comments: Eq. (13) => Eq. (5) (several times) After rebuttal: I thank the reviewers for their response.

Reviewer 3



This paper propose to adapt a method build surrogate losses based on adversarial formulation from binary classification to ordinal regression. A surrogate loss is proposed and is claimed to be consistent with the absolute error. Then, a set of experiments proves it to be competitive with the state-of-art algorithms of the task. - Presenting the 3 main theorems of the paper without any sketch of proof or simplified proof or intuition of proof in the paper bothered me. Make sure to add sketches of proofs in the final version. Content: - Despite the form, I'm convinced by the approach as it is an adaptation of the binary case. Which limits the impact of the paper in my opinion. - The paper proved the approach benefits from similar theoretical guarantees as state-of-art (such as Fisher consistency). - The experimental results show a competitive performance, without showing a significant improvement over the RED baseline.